# Effects of Dietary Anaplerotic and Ketogenic Energy Sources on Renal Fatty Acid Oxidation Induced by Clofibrate in Suckling Neonatal Pigs

**DOI:** 10.3390/ijms21030726

**Published:** 2020-01-22

**Authors:** Xi Lin, Brandon Pike, Jinan Zhao, Yu Fan, Yongwen Zhu, Yong Zhang, Feng Wang, Jack Odle

**Affiliations:** Laboratory of Developmental Nutrition, Department of Animal Sciences, North Carolina State University, Raleigh, NC 27695, USA; bepike2@ncsu.edu (B.P.); jnzhao@zinpro.com (J.Z.); fanyucau@163.com (Y.F.); zhuyw@scau.edu.cn (Y.Z.); yongzhang208@163.com (Y.Z.); fwang22@ncsu.edu (F.W.);

**Keywords:** Renal fatty acid oxidation, anaplerotic and ketogenic energy, PPARα activation

## Abstract

Maintaining an active fatty acid metabolism is important for renal growth, development, and health. We evaluated the effects of anaplerotic and ketogenic energy sources on fatty acid oxidation during stimulation with clofibrate, a pharmacologic peroxisome proliferator-activated receptor α (PPARα) agonist. Suckling newborn pigs (*n* = 72) were assigned into 8 dietary treatments following a 2 × 4 factorial design: ± clofibrate (0.35%) and diets containing 5% of either (1) glycerol-succinate (GlySuc), (2) tri-valerate (TriC5), (3) tri-hexanoate (TriC6), or (4) tri-2-methylpentanoate (Tri2MPA). Pigs were housed individually and fed the iso-caloric milk replacer diets for 5 d. Renal fatty acid oxidation was measured in vitro in fresh tissue homogenates using [1-^14^C]-labeled palmitic acid. The oxidation was 30% greater in pig received clofibrate and 25% greater (*p* < 0.05) in pigs fed the TriC6 diet compared to those fed diets with GlySuc, TriC5, and Tri2MPA. Addition of carnitine also stimulated the oxidation by twofold (*p* < 0.05). The effects of TriC6 and carnitine on palmitic acid oxidation were not altered by clofibrate stimulation. However, renal fatty acid composition was altered by clofibrate and Tri2MPA. In conclusion, modification of anaplerosis or ketogenesis via dietary substrates had no influence on in vitro renal palmitic acid oxidation induced by PPARα activation.

## 1. Introduction

The protective role of peroxisome proliferator-activated receptor α (PPARα) activation in nephrosis induced by fatty acids [1,2] and chronic kidney disease (CKD) [3] recently has been drawing great attention. The nephrosis and CKD can also be identified in neonates with an epigenetic origin [4], and physiological aAlterations in the bioactive lipids pattern [5] and oxidative status have been observed in the abnormal lipid metabolism [6]. Therefore, understanding renal development and adaptation of energy metabolism is very important for human infant health. Fatty acids are the preferred renal energy substrate, and it is believed that maintaining an active fatty acid metabolism is significant for the renal protective effects afforded by PPARα activation [2,7].

Activation of PPARα by pharmaceutical agonists such as clofibrate has been studied extensively in the liver, but not in kidneys. A few studies revealed that clofibrate feeding increases total and antimycin-insensitive oxidation rates in the kidneys of rats [8], and PPARα-mediated mechanisms are involved in gene expression and regulation of mitochondrial and peroxisomal β-oxidation enzymes induced by dietary lipids in the renal cortex of suckling rats [9]. This indicates that PPARα activation plays an important role in renal lipid metabolism. However, the enhancement of β-oxidation induced by clofibrate observed in rats usually is accompanied with peroxisome proliferation and hepatic carcinogenicity. Unlike rodent species, similar responses to clofibrate are not observed in the liver of humans or pigs [10,11]. Hence, the neonatal pig is commonly used as model for human pediatric nutrition and metabolism [12]. Recently we explored the effect of clofibrate on renal fatty acid oxidation in newborn pigs and confirmed that PPARα-mediated mechanisms are active [13]. Renal fatty acid oxidation in both mitochondria and peroxisomes was increased with oral administration of clofibrate (75 mg/kg body weight), and the natural developmental increase was greatly amplified [13]. The amplification induced by PPARα signaling may have great significance for preventing the development of the free fatty acid-associated renal disease. In addition, the amplification could promote the postnatal energy supply and thermogenesis that are essential for renal development and the survival of neonatal piglets. However, it is not known whether the high renal fatty acid β-oxidation could be impacted by modifying the dietary energy source. Studies with liver indicate that the fatty acid oxidative rate could be affected by the status of anaplerosis and ketogenesis. For example, addition of malic acid [14] and 2-methylpentanoate (2MPA, unpublished data) in liver homogenates from pigs and rats increased the fatty acid oxidation in vitro and modified the carbon distribution between CO_2_ and acid soluble products (ASP). The gene mitochondrial 3-hydroxy-3-methyl-glutaryl-CoA synthase (*mHMGCS*), coding for the key enzyme of ketogenesis is expressed significantly in liver and kidney after feeding clofibrate, but the ketone body levels in plasma and tissues remain low [13,15]. Because the excess acetyl-CoA generated from the elevated fatty acid β-oxidation needs to divert into the citric acid cycle (CAC) and the ketogenic pathway for further metabolism, the capacities of these pathways are critical for maintaining a high β-oxidative flux.

Renal energy metabolism is rather immature at birth, although fatty acids are the preferred energy substrate. The development of oxidative metabolism is progressive through the suckling period, in which PPARα activation is recognized as essential for fatty acid uptake, transport, and utilization. To maintain the efficacy of PPARα activation of renal lipid oxidative metabolism, in this study, we examined the roles of CAC anaplerosis and ketogenesis during elevated fatty acid oxidation by supplementation of extra anaplerotic carbon sources and ketogenic fatty acids such as medium-chain fatty acids (MCFA, [16]). MCFA were chosen because MCFA are absorbed quickly and oxidized in the mitochondria with no need of carnitine palmitoyl-transferase I (CPT I) transfer. Accordingly, MCFA are more ketogenic than long-chain fatty acids, and they have been used in ketogenic diets extensively. In addition, 2MPA (a branched-chain MCFA), valerate (an odd-chain MCFA), and hexanoate (an even-chain MCFA) were supplemented as triglycerides in the diets of this study. Among the MCFA, 2-MPA is a novel source of anaplerotic carbon via the propionyl-CoA/succinyl-CoA pathway, valerate is a dual source of both anaplerotic and ketogenic carbon via producing propionyl-CoA and acetyl-CoA and hexanoate is ketogenic via production of acetyl-CoA and acetoacetyl-CoA. Furthermore, succinate, an intermediate of the CAC, was used as a direct anaplerotic carbon source due to its potential inactivation of mHMGCS via succinyl-CoA [17]. We hypothesized that modification of the capacities of the CAC and ketogenic pathways affects the elevated fatty acid oxidative flux induced by supplementation of clofibrate in milk (0.35% of dry matter) and alters the distribution of the fatty acid oxidative metabolites for renal energy generation. The data obtained from this study would be useful in understanding the role of dietary energy source in fatty acid oxidation induced by PPARα activation in the kidney during early development.

## 2. Results

### 2.1. Animal Growth Performance

There were no effects of supplementation of clofibrate or dietary addition of 5% glycerol + succinate (GlySuc), triglycerides of valeric acid (TriC5), hexanoic acid (TriC6), and 2MPA (Tri2MPA) on piglet growth performance (*p* > 0.1). The overall piglet daily gains, liquid diet intakes, and final body weights were 120 ± 11 g, 780 ± 23 g, and 2.0 ± 0.07 kg, respectively. Supplementation of clofibrate into the milk replacer also had no effects on kidney weight or protein concentration. The average kidney weight was 8.1 ± 0.3 g, and kidney protein concentration was 33.9 ± 1.4 mg /g of fresh tissue.

### 2.2. Renal Fatty Acid Profile

No interaction was observed between clofibrate and dietary treatment for each fatty acid (*p* > 0.05). The main effects of clofibrate and the dietary treatment were reported in Table 1. Dietary supplementation of clofibrate increased renal fatty acid concentrations (µg/100 mg tissue) of C16:1 n9, C18:1 n9 (*p* < 0.04) and C18:3 n3 and C18:3 n6 (*p* < 0.01) but decreased in C22:6 n3 (*p* < 0.01) (Table 1). Addition of Tri2MPA to the diet increased concentrations of C15:0, but decreased C18:2 n6, C20:2 n7, and C20:3 n6 (*p* < 0.02) compared with other dietary additions (GlySuc, TriC5, and TriC6). Generally, clofibrate increased mono fatty acids and dietary Tri2MPA decreased polyunsaturated fatty acids (*p* < 0.05). Clofibrate had no impact on the total fatty acid, while dietary Tri2MPA reduced the total fatty acids. No clofibrate or dietary treatment had an impact on n6/n3 fatty acid ratio.

### 2.3. Renal β-hydroxybutyrate and Acetate Concentrations

Supplementation of clofibrate had no detectable effect on renal β-hydroxybutyrate (*p* = 0.1) or acetate concentrations. However, the concentration of β-hydroxybutyrate was higher in kidney tissue from pigs fed Tri2MPA and TriC6 than in controls fed GlySuc (*p* < 0.02). No interactions between clofibrate and other dietary treatments (*p* = 0.6) were detected in β-hydroxybutyrate. There were no effects of clofibrate supplementation or dietary energy treatments on the renal acetate concentration, but the concentration of acetate was 112-fold higher than β-hydroxybutyrate (Table 2).

### 2.4. Renal Palmitic Acid Oxidation

No interactions were detected between clofibrate and other dietary treatments (GlySuc, TriC5, TriC6, and Tri2MPA) for palmitic acid oxidation (*p* ≥ 0.1). Supplementation of clofibrate in the milk replacer increased palmitic acid oxidation (*p* < 0.0001). The ^14^C accumulations (nmol/(h.mg fresh tissue protein)) in CO_2_, ASP, and CO_2_ + ASP were 31, 28, and 30% higher in kidney from pigs fed the diets with clofibrate than without clofibrate (Figure 1).

Supplementation of TriC6 in the milk replacer also increased the palmitic acid oxidation (*p* < 0.001). The ^14^C accumulations (nmol/(h.mg fresh tissue protein)) on average in CO_2_, ASP, and CO_2_ + ASP were 25%, 24%, and 25% higher in kidney from pigs fed the diet with TriC6 than the diets with GlySuc, TriC5, and Tri2MPA (Figure 2).

There were no interactions between clofibrate/dietary treatment and the factors (carnitine, iodoacetamide, and L659699) added in vitro incubation medium (*p* > 0.47), and the effects of the factors on palmitic acid oxidation were reported in Figure 3. Addition of carnitine to the oxidation medium in vitro increased the renal palmitic acid oxidation (*p* < 0.0001). The ^14^C accumulations (nmol/(h.mg fresh tissue protein)) on average in CO_2_, ASP, and CO_2_ + ASP were 74%, 128%, and 102% greater from carnitine than control and L659699 or iodoacetamide, the inhibitors of ketogenesis (Figure 3A). Supplementation of carnitine also changed the distribution (%) between CO_2_ and ASP, reducing ^14^C accumulation in CO_2_ by 16% and increasing that in ASP by 13% (Figure 3B). Neither L659699 nor iodoacetamide had an effect on oxidation (*p* > 0.1). Additionally, the ^14^C accumulations (nmol/(h.mg fresh tissue protein)) in CO_2_, ASP, and CO_2_ + ASP from 5-day old pigs were greater than newborn pigs (*p* < 0.001), regardless of treatment. However, age did not alter the distribution between CO_2_ and ASP (data not shown).

### 2.5. Enzyme Activity

There was an interaction between clofibrate and dietary treatments (*p* < 0.002) for citrate synthase activity (Figure 4A). Supplementation of clofibrate increased the citrate synthase activity in pigs fed GlySuc (*p* < 0.004), but no difference was detected among all other dietary treatment groups (*p* ≥ 0.1). The activity of citrate synthase was on average 2.5-fold higher from 5 d-old pigs than newborn pigs. An interaction between clofibrate and the dietary treatment (*p* < 0.04) was also observed for propionyl-CoA carboxylase (Figure 4B).

Supplementation of clofibrate had no significant influence on the propionyl-CoA carboxylase activity (*p* = 0.1), but increased the enzyme activity in pigs fed TriC6 compared to pigs fed GlySuc (*p* < 0.02), and decreased activity in pigs fed Tri2MPA compared to those fed TriC6 (*p* < 0.02). The activity of propionyl-CoA carboxylase (Figure 4B) was on average 1.5-fold higher from 5 d-old pigs than newborn pigs (*p* < 0.0075).

No interaction between clofibrate and the dietary treatment was detected (*p* > 0.05) for acetyl-CoA carboxylase (Figure 5). Supplementation of neither clofibrate nor the MCFA had an effect on the activity of acetyl-CoA carboxylase (*p* = 0.09). The activity of acetyl-CoA carboxylase (Figure 5) was on average 3.7-fold greater from 5 d-old pigs than newborn pigs (*p* < 0.04).

### 2.6. Gene Expression

There were no effects of clofibrate and dietary treatments on the abundance of genes associated with renal fatty acid metabolism (*PPARα*, *MCD*, *KCoA*, *PGC1α*, *FGF21*, *MCAD*, *LCAD*, and *VLCAD*) measured in this study (*p* ≥ 0.11; Appendix A).

## 3. Discussion

Limited studies have examined the developmental aspects of renal fatty acid oxidation. Palmitic acid oxidation by neonatal kidney from rats increased after birth and reached a maximum between 5 and 7 days of postnatal life [18]. Consistent with these results, palmitic acid oxidation in our study showed a 1.9-fold higher rate in 5 d-old pigs than in newborn pigs. Moreover, the developmental increase could be amplified further by addition of clofibrate into the milk diet, without altering kidney size (weight) or growth performance of the piglets. Similar results were observed from sow-raised pigs receiving an oral gavage of clofibrate in our previous study [13]. In that study, the amplified β-oxidation was associated with an induction of gene expression and activity of carnitine palmitoyl-transferase I (CPTI), one of the target genes of PPARα [13]. The increased CPTI activity was accompanied by a rapid reduction in the sensitivity of CPTI and an increase in the gene expression of malonyl-CoA decarboxylase (*MCD*) in the liver of swine [15], suggesting that malonyl-CoA is involved in the regulation of fatty acid catabolism induced by clofibrate. Unlike in the liver, however, the effects of clofibrate on *MCD* was not detected in the kidneys in this study. In addition to CPTI, the effects of clofibrate on expression of genes *PGC1α*, *MCAD*, *LCAD*, and *VLCAD* with increased PPARα were reported in the nephrotic kidney [2] in rats. However, the expressions of these genes were not significantly different between pigs with and without administration of clofibrate in this study. The physiological status and response of species to clofibrate administration could be factors for the variations, and apparently need to be further investigated.

The induction of renal fatty acid oxidation via activation of PPARα occurred with the same pattern as observed in the liver of neonatal pigs [19]. Although renal β-oxidation was increased as previously observed [13], the increase had a minimal impact on ketogenesis probably due to a limited activity of HMGCS, owing to a posttranscriptional defect described in pigs [20]. Because of the diminished ketogenesis observed in neonatal pigs [21], we have been interested in the effects of modifying CAC anaplerosis and ketogenesis on fatty acid utilization, especially when β-oxidation is induced by PPARα activation. In this study, hexanoic acid was used as a source of ketogenic carbon for both mHMGCS and acetoacetyl-CoA deacylase (AACD) pathways via acetyl-CoA or/and acetoacetyl-CoA. In contrast, 2MPA was selected as a novel source of anaplerotic carbon via the propionyl-CoA-succinyl-CoA pathway and valeric acid was used as a dual source of both anaplerotic and ketogenic carbon via its production of both propionyl-CoA and acetyl-CoA from β-oxidation. The CAC intermediate, succinate, was used as a direct anaplerotic carbon source and also due to its role in inhibition of ketogenesis by inactivation of mHMGCS via succinyl-CoA [17]. Although the activity of mHMGCS could not be improved by induction of the gene expression via activation of PPARα [15], we expected that ketone bodies might be produced via the AACD pathway when fatty acid β-oxidation is increased. Indeed, feeding diets containing MCFA increased β-hydroxybutyrate concentration as compared to succinate in kidney tissues, suggesting that the renal ketogenic pathways could be modified by diet. However, the modification was not associated with clofibrate because no interaction with dietary energy sources was detected. Furthermore, the modification of ketogenesis by dietary MCFA might not be due to a change in CAC cataplerosis activity, because there was no difference between the even, odd, and branched-MCFA. Thus, the increase in renal β-hydroxybutyrate from dietary MCFA could be due to an increase in citrate synthase and a relative decrease in β-hydroxybutyrate from dietary succinate, in which the mHMGCS could be inactivated by succinyl-CoA derived from succinate [17].

Although dietary TriC6 and Tri2MPA increased renal β-hydroxybutyrate as compared to the diet containing succinate, palmitic acid oxidation increased only in pigs that received the diet containing TriC6. No evidence of anaplerotic effects on β-oxidation were observed in pigs fed GlySuc, TriC5, and Tri2MPA treatments. Furthermore, the enzyme activity of propionyl-CoA carboxylase was unaltered by these treatments. The interaction between anaplerotic and cataplerotic reactions was studied in human kidney during prolonged starvation [22]. A balance of anaplerotic reactions to replenish the α-ketoglutarate in the CAC and cataplerotic reactions to drain remnant 4-carbon metabolic intermediates from the cycle for glucose synthesis is essential and required of the renal tissue during starvation [23,24]. This suggests that entry and removal of intermediates into or out of the CAC, as in the fasted kidney, were likely balanced in our study. Thus, the effect of anaplerotic carbon on fatty acid oxidation might not be detected in vitro in the limited tissue homogenates of our study. However, the stimulatory impact of TriC6 on palmitic acid oxidation was unexpected. We speculate that it could be due to the regulatory role of acetyl-CoA (as a positive allosteric regulator) in the archetypical anaplerotic enzyme, pyruvate carboxylase [24], and potentially a more available free CoA pool in the kidney tissues from piglets fed diet with TriC6. Compared with propionyl-CoA, acetyl-CoA releases CoA and acetate via acetyl-CoA deacylase, while reversibility of methylmalonyl-CoA generated from propionyl-CoA would be reduced when propionyl-CoA is increased [25]. In addition, supplementation of inhibitors of the ketogenesis pathways had no effect on the palmitic acid oxidation and its oxidative metabolite distribution between CO_2_ and ASP, suggesting that renal ketogenesis (similar to piglet liver) is negligible from either mHMGCS or AACD. In support of this, the acetate measured in kidney was over 100 times higher than β-hydroxybutyrate. This is consistent with our previous findings in liver, with acetate being the predominant metabolic product of fatty acid oxidation [14] in neonatal pigs. The physiological role of acetate in mitochondria has not been stressed to an adequate extent, however, the CoA required for maintaining β-oxidation rate would be released by generating acetate from the acetyl-CoA. An increase in acetate has been observed in vitro under conditions of high fatty acid oxidation [14].

Feeding clofibrate had a profound impact on cardiac fatty acid composition in rats [26], but the impact on renal fatty acid composition has not been studied in pigs, rodents, or any other species. Clofibrate significantly reduced the n6/n3 fatty acid ratio in the hearts of rats by decreasing linoleic acid (C18:2 n6) and arachidonic acid (C20:4 n6), which was highly associated with the animals’ lipid peroxidation status [26]. We found that clofibrate significantly increased total monounsaturated fatty acid and α- and γ-Linolenic acid (C18:3 n3 and n6) and reduced docosahexaenoic acid (C22:6 n3). However, the alternations of these fatty acids had no impact on the n6/n3 ratio, suggesting that effect of clofibrate on fatty acid composition might vary between tissues. Because dietary fat type (especially medium-chain triglycerides (MCT)) affects tissue fatty acid composition, feeding diets with high levels of MCT had lower n3 and greater n6 fatty acid levels in the neutral lipid fraction of muscle tissue [27]. Therefore, the variations in fatty acid composition might not be impacted by clofibrate only but also by diet composition. Indeed, the total fatty acid and polyunsaturated fatty acids, especially the n6 fatty acids, were significantly reduced in pigs who received the Tri2MPA diet in this study. This is a very interesting finding and could be due to the role of 2MPA in CAC anaplerosis. It has been reported that the monomethyl branched-chain fatty acids could be accumulated in the liver by feeding a high fat diet with protein [28], implying that 2MPA, as an anaplerotic substrate, plays a role in fatty acid metabolism.

It is remarkable that the addition of carnitine in the incubation medium significantly increased palmitic acid oxidation. Moreover, the increase in ^14^C accumulation in ASP was much greater than in CO_2_, resulting in a significant change in the distribution between CO_2_ and ASP. These results indicate that the carnitine level in the collected kidney tissues could be a limiting factor for palmitic acid oxidation. Tissue carnitine level is closely related to the dietary carnitine intake and potential capacity of endogenous synthesis in mammals [29,30]. However, the contribution of carnitine from endogenous synthesis to carnitine status is not known due to a lack of data and appropriate animal models [31]. In addition to dietary intake and endogenous synthesis, the carnitine status of neonatal pigs can be affected by the maternal carnitine status. Increased tissue free and total carnitine concentrations was observed in the fetuses collected from gilts given dietary carnitine supplementation [32]. Thus, in the present study, deficient carnitine in the kidney tissues collected might be related to the carnitine levels in the milk replacer and potential carnitine status at birth. Furthermore, the high percentage of ASP noticed in in vitro palmitic acid oxidation suggested that carnitine availability is essential for maintaining an active lipid metabolism, because the potential toxic coenzyme A esters can be removed by forming acylcarnitines, resulting in an increase in percentage of ASP during the fatty acid oxidation. The buffer function of carnitine in the cellular acyl-coenzyme A/coenzyme A ratio plays a critical role in normal metabolic functioning [33,34].

## 4. Materials and Methods

### 4.1. Animals

A total of 72 suckled newborn pigs (male and females removed from their mothers within 6 h after birth, average body weight 1.31 ± 0.027 kg) from eight litters were used in this experiment. Eight pigs (one from each litter) were euthanized at birth by being sedated with isoflurane followed by exsanguination. The kidneys were removed, and weight was recorded immediately. Kidney tissue samples were collected and stored at −80 °C for subsequent analyses. All other pigs were housed individually in a specialized intensive care nursery at 30 °C. Pigs were blocked by litter and assigned randomly into eight treatments according to a 2 (control vs. clofibrate) × 4 (diets supplemented with either succinate, valerate, hexanoate, or 2MPA) factorial design. Each group was fed a basal milk replacer containing either 0 or 0.35% of the clofibrate (% of the milk dry matter) with the supplementations of 5% glycerol + succinate (GlySuc), triglycerides of valeric acid (TriC5), hexanoic acid (TriC6), or 2-MPA (Tri2MPA). The basal milk replacer was formulated with soybean oil as a fat source with a total of 25% fat, 31% crude protein, and 36% lactose. The formulated basal milk replacer contained 12.8% of dry matter. Based on our previous study [15], the average clofibrate intake would be ~200 mg/kg body weight per day if the average milk intake was 750 mL and the average body weight was 1.65 kg throughout the experimental period. Supplements to the basal diet were offset by reductions in soybean oil so that diets remained iso-caloric. The TriC5, TriC6, and Tri2MPA were synthesized in our laboratory as previously described [35]. A gravity feeding system was used to allow for the accurate measurement of milk consumption [36]. Fresh milk was provided three times per day, and diets were prepared at the arrival of each litter and stored under refrigeration. Body weight was recorded daily. Pigs were euthanized as newborn pigs on day 5 and tissues were collected and stored as previously described. All experimental procedures were approved by the North Carolina State University animal care and use committee (IACUC id 16-142, approved on 14 September 2016).

### 4.2. Fatty Acid Oxidation Measurement In Vitro

Palmitate oxidation was measured in fresh kidney homogenates as previously described [15]. Measurements were conducted using [1-^14^C] palmitic acid (0.5 mM, 0.28 kBq/µmol) pre-incubated with or without carnitine (1 mM), L659699 (1.6 µM), an inhibitor of both mitochondrial HMGCS and cytosolic HMGCS [37], or iodoacetamide (50 µM), an inhibitor of acetoacetyl-CoA deacylase (AACD) for acetoacetate synthesis [38]. Oxidative metabolites (^14^CO_2_ and ^14^C-ASP) were determined using liquid scintillation counting [14].

### 4.3. Metabolite Assays

Fatty acids were methylated using the method described by Walter et al. [39], and the fatty acid methyl esters were quantified using GC/MS (Agilent Technologies, Wilmington, DE, USA). Ketone bodies were determined using EnzyChromTM Ketone Body Assay Kit (EKBD-100) purchased from BioAssay Systems (Hayward, CA, USA), and acetate was determined using an acetate colorimetric kit from BioVision (Milpitas, CA, USA).

### 4.4. Enzyme Assays

Citrate synthase activity in mitochondria was measured with a kit purchased from Sigma-Aldrich (MAK193; St. Louis, MO, USA). The activities of acetyl-CoA and propionyl-CoA carboxylases were measured following the method described by Hugler et al. [40] with a slight modification. Briefly, homogenates from frozen tissues (0.5–0.65 mg protein) were incubated with acetyl-CoA or propionyl-CoA (0.4 mM) and [^14^C]Na_2_CO_3_ (37 kBq/µmol) in a buffer containing 100 mM Tris/HCl, pH 7.8, 5 mM MgCl_2_, 5 mM dithioerythritol, 4 mM ATP, 2 mM NADPH, and 10 mM NaHCO_3_ at 37 °C for 6 min. The reaction was started by adding the substrate (acetyl-CoA or propionyl-CoA) and ended by adding HCl. The unused substrate then was removed from the reactant using an analytical evaporator (MULTVAP, Berlin, MA, USA) under N_2_.

### 4.5. Gene Expression

To test the effect of clofibrate on fatty acid metabolism in neonatal kidney, kidney samples were snap frozen in liquid nitrogen for analyzing mRNA abundance of peroxisome proliferator-activated receptor α (*PPARα*), malonyl-CoA decarboxylase (*MCD*), 3-ketoacyl-CoA thiolase (*K-CoA*), fibroblast growth factor 21 (*FGF21*), peroxisome proliferator-activated receptor γ coactivator-1α (*PGC1α*), medium-chain acyl-CoA dehydrogenase (*MCAD*) and long-chain acyl-CoA dehydrogenase (*LCAD*), and very long-chain acyl-CoA dehydrogenase (*VLCAD*) via qPCR. Guanidine isothiocynate was used for total RNA extraction. The extracted mRNA was quantified with spectrophotometry (NanoDrop, Thermo Scientific, Wilmington, DE, USA) and treated with Turbo DNase (Ambion, Austin, TX, USA). The treated mRNA then was transcribed using an iScripTM Select cDNA synthesis kit (Bio-Rad Laboratories, Hercules, CA, USA). Primers were designed based on the use of GenBank, as described previously (Appendix A, [14]). The mRNA abundances were measured with MyiQ Single Color RT-PCR (Bio-Rad Laboratories, Hercules, CA, USA).

### 4.6. Chemicals

Palmitic acid [1-^14^C] and sodium bicarbonate [^14^C] were purchased from American Radiolabeled Chemicals, Inc. (Saint Louis, MO, USA). L659699 was purchased from Cayman Chemical (Ann Arbor, MI 48108, USA), clofibrate was purchased from TCI America (Portland, OR, USA), and all other chemicals were purchased from Sigma-Aldrich (Saint Louis, MO, USA).

### 4.7. Statistical Analysis

Data from enzyme assays was subjected to ANOVA according to a 2 × 4 factorial randomized complete block design using the general linear models (GLM) procedure of SAS (SAS software 9.3, Cary, NC, USA). Data from in vitro fatty acid oxidative measurements of 5 d-old pigs were subjected to ANOVA according to split-plot design, using the GLM procedure of SAS. The main plot was the eight dietary treatments on animals and the sub-plot was the four treatments on the tissues (with or without carnitine, L659699, and iodoacetamide). Data from the 5 d-old pigs were compared to the data from newborns also using ANOVA according a randomized completed design with pre-planned contrasts. Animal replication was projected using power tests employing data (means and stdev) from our previous studies [14,41] and the power for two-tail at >0.9. When the interactions between clofibrate and dietary medium-chain fatty acid treatments (main dietary factors) or the main dietary factors and in vitro treatments were not statistically significant, we presented the results based on the significant effects of the main factors only. All values are presented as least squares means ± SEM. Differences were declared at a *p*-value of ≤ 0.05.

## 5. Conclusions

Clofibrate induces fatty acid oxidation and alters renal fatty acid composition. Dietary supplementation of MCFA as anaplerotic and ketogenic carbon sources may modify the metabolic pathways of renal fatty acid oxidation, but the modification is minimal and has no effect on the elevated fatty acid oxidation induced by activation of PPARα at current status. The renal ketogenic capacity is low in piglets, and the simulative effect from TriC6 may be due to acetogenesis, the predominate pathway for releasing the CoA from acetyl-CoA generated from β-oxidation. The role of carnitine in kidney is essential and merits further investigation in neonatal pigs.

## Figures and Tables

**Figure 1 ijms-21-00726-f001:**
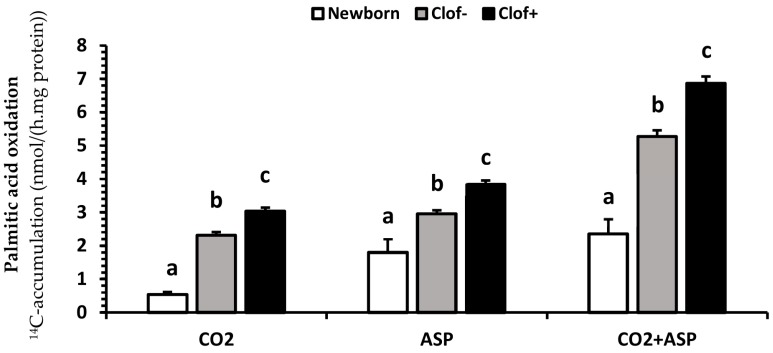
The main effect of clofibrate on palmitic acid oxidation. Data (Lsmeans ± SEM) are ^14^C-accumulations in CO_2_, acid soluble products (ASP) and CO_2_ + ASP (*n* = 32). ^abc^ Treatments (newborn, clof− (no clofibrate supplementation) and clof+ (clofibrate supplementation)) for CO_2_, ASP and CO_2_ + ASP with different superscripts differ (*p* < 0.05).

**Figure 2 ijms-21-00726-f002:**
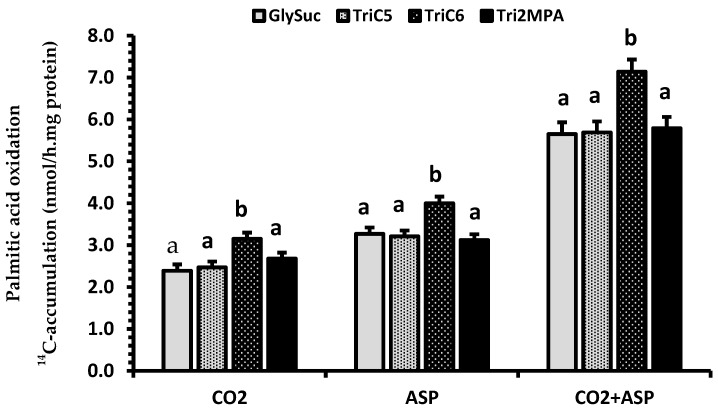
The main effect of dietary treatments on [1-^14^C]-palmitic acid oxidation. GlySuc (diet supplement with glycerol and succinate), TriC5 (triglyceride of valerate), TriC6 (triglyceride of hexanoate), and Tri2MPA (triglyceride of 2MPA). Data (Lsmeans ± SEM) are ^14^C-accumulations in CO_2_, acid soluble products (ASP) and CO_2_ + ASP (*n* = 16). ^a,b^ Treatments for CO_2_, ASP, and CO_2_ + ASP with different superscripts differ (*p* < 0.05).

**Figure 3 ijms-21-00726-f003:**
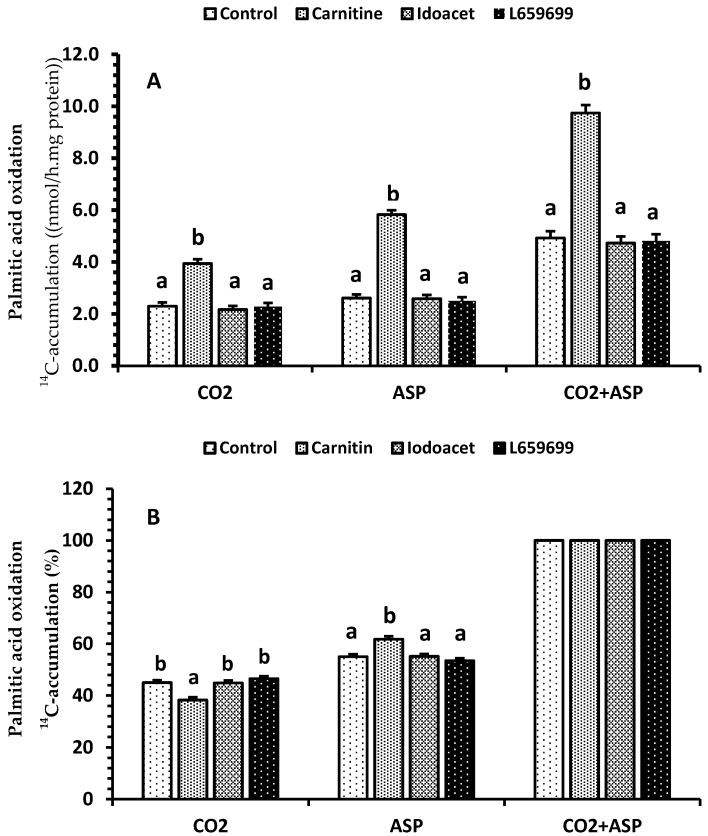
The main effect of carnitine and inhibitors of 3-hydroxy-3-methyl-glutaryl-CoA synthase (HMGCS) and acetoacetyl-CoA deacylase (AACD) on palmitic acid oxidation. Fresh kidney homogenates were incubated with palmitic acid only (control), and carnitine, iodoacetamide (Idoacet), or L659699. Data in A (Lsmeans ± SEM) are ^14^C-accumulations in CO_2_, acid soluble products (ASP), and CO_2_ + ASP (*n* = 16). Data in B (Lsmeans± SEM) are the percentage of the ^14^C-accumulations in CO_2_, acid soluble products (ASP), and CO_2_ + ASP (*n* = 16). ^a,b^ Treatments in (**A**) and (**B**) for CO_2_, ASP, and CO_2_ + ASP with different superscripts differ (*p* < 0.05).

**Figure 4 ijms-21-00726-f004:**
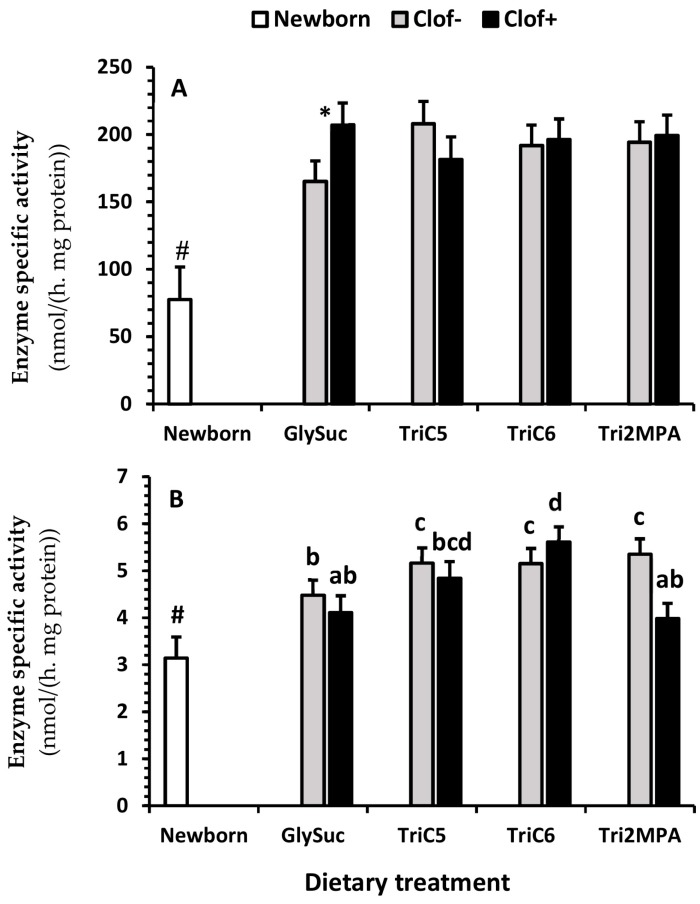
The effect of clofibrate and dietary treatments on citrate synthase and propionyl-CoA carboxylase activity. Data (Lsmeans ± SEM) are specific activities (nmol/(h.mg of tissue protein)); *n* = 16). * indicates difference in (**A**) between pigs fed GlySuc with and without clofibrate (*p* < 0.004). ^a,b,c,d^ Treatments cross columns in (**B**) with different superscripts differ (*p* < 0.05). ^#^ indicate difference in (**A**,**B**) between newborn and 5 d-old pigs (*p* < 0.05).

**Figure 5 ijms-21-00726-f005:**
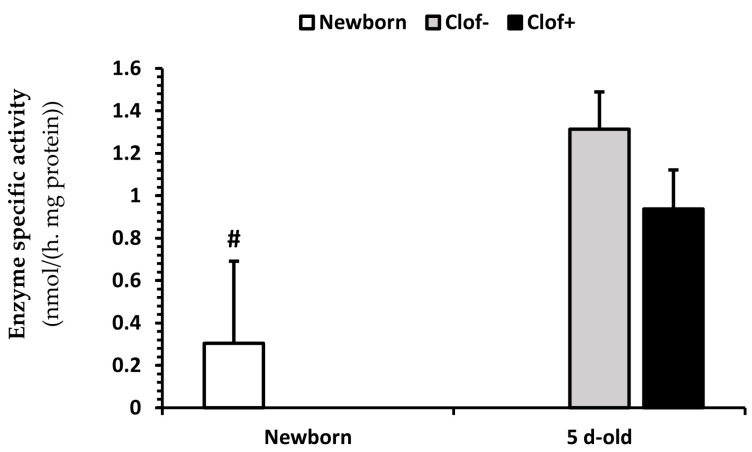
The effect of dietary clofibrate fed to pigs for five days on acetyl-CoA carboxylase activity. Data (Lsmeans ± SE) are specific activities (nmol/(h.mg of tissue protein)). No significant difference was observed between pigs with and without clofibrate (*p* = 0.06). ^#^ The enzyme activity was on average 3.7-fold greater from 5 d-old pigs than newborn pigs (*p* < 0.04).

**Table 1 ijms-21-00726-t001:** Effect of clofibrate and medium-chain triglycerides on renal fatty acid composition.

	Main Effects ^@^											
	Clofibrate					Medium-Chain Fatty Acid					
FA	NB ^†^	Clof−	Clo+	SEM	*p*-V	GlySuc	TriC5	TriC6	Tri2MPA	SEM	*p*-V
		*µg/100 mg Tissue*									
C14:0	2.49	0.46	0.89	0.18	0.087	0.48	0.81	0.90	0.52	0.25	0.538
C15:0	1.26	0.25	0.37	0.06	0.201	0.09 ^a^	0.38 ^b^	0.11 ^a^	0.68 ^c^	0.08	0.001
C16:0	149	122	123	47.0	0.742	127	124	132	108	6.56	0.081
C16:1n9	14.5	2.79	4.72 *	0.50	0.001	3.17	4.41	3.94	3.40	0.71	0.605
C17:1	0.68	0.38	0.46	0.10	0.637	0.23	0.39	0.42	0.60	0.14	0.288
C18:0	93.3	102	95.5	3.19	0.834	99.3	99.5	103	94.0	4.59	0.533
C18:1n9	175	113	136 *	7.00	0.041	102	125	137	132	9.84	0.087
C18:2n6	64.2	117	132	6.39	0.101	140 ^b^	127 ^b^	136 ^b^	92.5 ^a^	8.96	0.002
C18:3n6	0.63	0.46	0.71 *	0.06	0.005	0.53	0.58	0.49	0.73	0.09	0.228
C18:3n3	0.36	2.20	3.06 *	0.23	0.013	3.16	2.82	2.07	2.48	0.34	0.124
C20:0	4.46	3.91	3.74	0.19	0.570	3.88	3.80	3.80	3.85	0.28	0.995
C20:1n9	1.98	0.83	0.96	0.06	0.190	1.02	0.98	0.86	0.71	0.09	0.091
C20:2n7	6.90	6.02	6.58	0.45	0.481	7.76 ^b^	6.63 ^ab^	6.17 ^ab^	4.84 ^a^	0.07	0.022
C20:3n6	6.81	4.65	5.54	0.35	0.091	4.70 ^ab^	5.88 ^b^	5.78 ^b^	3.97 ^a^	0.06	0.028
C20:4n6	155	120	106	5.55	0.088	115	120	117	100	8.12	0.303
C20:3n3	0.48	3.20	2.50	0.48	0.168	2.99	2.45	3.86	2.09	0.68	0.292
C20:5n3	1.28	1.52	1.75	0.11	0.223	1.53	1.68	1.80	1.47	0.16	0.492
C22:0	3.51	4.20	3.67	0.20	0.068	4.10	4.02	3.97	3.67	0.28	0.953
C22:1n9	0.30	0.26	0.30	0.07	0.709	0.31	0.38	0.22	0.19	0.09	0.516
C22:2	0.29	0.36	0.43	0.07	0.508	0.55	0.36	0.21	0.46	0.09	0.079
C23:0	1.62	1.50	1.76	0.15	0.298	1.46	1.50	1.62	1.88	0.24	0.494
C22:5n3	11.8	7.94	8.54	0.55	0.388	8.83	8.30	8.08	7.36	0.78	0.342
C24:0	4.43	5.74	5.58	0.28	0.662	6.14 ^b^	5.85 ^b^	6.03 ^b^	4.56 ^a^	0.39	0.022
C22:6n3	8.35	10.6	8.13 *	0.66	0.009	9.94	10.0	9.55	7.87	0.93	0.324
C24:1	10.3	8.89	8.10	0.54	0.260	8.84	9.22	8.80	6.95	0.77	0.164
Sum	789	697	722	21.1	0.424	731 ^b^	732 ^b^	759 ^b^	617 ^a^	30.3	0.009
MUFA	203	127	149 *	7.46	0.043	139	151	145	114	10.8	0.080
PUFA	255	275	274	11.3	0.964	296 ^b^	286 ^b^	291 ^b^	223 ^a^	16.6	0.009
FAn3	21.2	25.6	23.7	1.56	0.774	26.5	25.3	25.4	21.3	2.24	0.343
FAn6	227	243	243	10.1	0.971	260 ^b^	253 ^b^	259 ^b^	197 ^a^	14.6	0.008
Fan6/n3	11.5	10.5	10.9	0.55	0.556	10.6	10.9	11.2	10.1	0.81	0.447

^@^ No interaction was detected between clofibrate and dietary medium-chain fatty acid effects. Tabulated data represent least square means and standard error of the means for the main effects (clofibrate effect: *n* = 32; medium-chain fatty acid effect: *n* = 16). * Least square means under the main effect of clofibrate within a row differ (*p* < 0.05); Clof−: all pigs received milk with no clofibrate supplementation and Clof+: all pigs received milk with clofibrate supplementation. ^a,b,c^ Least square means under the main effect of medium-chain fatty acid within a row lacking a common superscript differ (*p* < 0.05); GlySuc: glycine + succinate, TriC5: triglyceride of valerate, TriC6: triglyceride of hexanoate and Tri2MPA: triglyceride of 2-methylpentanoate. ^†^ Newborn (NB) concentrations of C14:0, C15:0, C16:1n9, C20:5n3, C20:1n9 C18:3n3, and C20:3n3 are different from 5-d old pigs (*p* < 0.05).

**Table 2 ijms-21-00726-t002:** Concentrations of β-hydroxybutyrate and acetate in kidney tissue.

	β-Hydroxybutyrate		Acetate	
Main Effects ^@^	LSmean	SEM	LSmean	SEM
	µmol/g Tissue		µmol/g Tissue	
*Clofibrate* ^†^				
Clof−	1.68	0.12	197	23.8
Clof+	1.94	0.12	210	24.7
*p*-Value	0.13		0.70	
*Dietary medium-chain fatty acid* *				
GlySuc	1.42 ^a^	0.17	194	35.1
TriC5	1.78 ^ab^	0.17	224	37.5
TriC6	1.89 ^b^	0.16	204	32.1
Tri2MPA	2.14 ^b^	0.16	192	32.1
*p*-Value	0.023		0.91	

^@^ No interaction was detected between clofibrate and dietary medium-chain fatty acid. The main effects of clofibrate and dietary medium-chain fatty acid were reported. ^†^ The main effect of clofibrate (*n* = 28), Clof−: all pigs received milk with no clofibrate supplementation and Clof+: all pigs received milk with clofibrate supplementation. * The main effect of dietary medium-chain fatty acid (*n* = 16). GlySuc: glycine + succinate, TriC5: triglyceride of valerate, TriC6: triglyceride of hexanoate and Tri2MPA: triglyceride of 2-methylpentanoate. Tabulated data represent least square means (LSmean) and standard error of means (SEM). ^a,b^ Least square means under the main effect within column lacking a common superscript differ (*p* < 0.05).

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
