# Peer review of "Effects of Dietary Anaplerotic and Ketogenic Energy Sources on Renal Fatty Acid Oxidation Induced by Clofibrate in Suckling Neonatal Pigs"

_ijms, 2020, doi:10.3390/ijms21030726_

Round 1
Reviewer 1 Report
Manuscript ID ijms-664807 entitled "Effects of dietary anaplerotic and ketogenic energy sources on renal fatty acid oxidation induced by clofibrate in suckling neonatal pigs" by Lin X. et al, reports the effects of anaplerotic and ketogenic diet on fatty acid oxidation at renal level during treatment with clofibrate. In my opinion the manuscript is quite complete and results are clearly presented. I only suggest to summarize the discussion section that, in the present form, is too long while it should be more coincise.
Author Response
Manuscript ID ijms-664807 entitled "Effects of dietary anaplerotic and ketogenic energy sources on renal fatty acid oxidation induced by clofibrate in suckling neonatal pigs" by Lin X. et al, reports the effects of anaplerotic and ketogenic diet on fatty acid oxidation at renal level during treatment with clofibrate. In my opinion the manuscript is quite complete and results are clearly presented. I only suggest to summarize the discussion section that, in the present form, is too long while it should be more coincise.
Response: We have shortened the discussion by 5% and we have shortened the conclusions by 33%. Thank you for the suggestion.
Reviewer 2 Report
1) We don't know whether te study was approved by an ethical committee
2) Piglets were all weaned from birth ? What did they receive as feed ?
3) the paper needs to be rewitten, specially the results section. I do not understand your tables.
In table 1 plesae include in the lengend what clo, clo, mean.
I don’t understand table 1, for instance was is the result for C14 :0 with chlor alone, vs chloro+Glysuc, vs no chloro + glysuc. The first p-V is the results of which comparison ? as 2.4 for NB could be statistically different from 0.46 with Clof-, and 0.89 with Clo+. Same question for the second part of the line.
Table 2 : Glysuc is glysuc and clo+ or clo - ? same for the other results
Figure 2 : indicate whether feed supplementation was done with or without clo. From my point of view there should be 8 conditions, i.e for each feed supplementation +/- clo. So when not performed so, it should be cleraly stated, justified and the results takenn with very cautionously in the discussion section.
Same remark for figure 3 and 4. I don’t understand all letters in graph b in figure 4 : who is who in the bar ? I suppose that A is citrate synthase activity and b propionylCoA carboxylase one ? Informations of figure 5 should be in figure 4. In figure 5 SEM are important so that it is not obvious that white bar is statistically different for the black one. There is no tendency in science : either it is statistically significant, either it is not.
The discussion will have to be modified after that the missing elements in the results section have been included.
Author Response
1) We don't know whether the study was approved by an ethical committee
Response 1: The study was approved by our university ethical committee (page 11, line 342). The IACUC id is 16-142 and was approved on September 14, 2016). We have added the ID number and approved date in the revised manuscript.
2) Piglets were all weaned from birth ? What did they receive as feed ?
Response 2: All pigs in this study (Page 10, line 325) were weaned within 6 hours of birth, and received mom’s milk (suckled) as feed before they were removed from their moms. We have clarified this in the revised manuscript
3) the paper needs to be rewitten, specially the results section. I do not understand your tables. In table 1 plesae include in the lengend what clo, clo, mean.
Response 3 for point 1: The definition of the abbreviations (clo- and clo+) have been included in the table legend in the revised manuscript.
I don’t understand table 1, for instance was is the result for C14 :0 with chlor alone, vs chloro+Glysuc, vs no chloro + glysuc. The first p-V is the results of which comparison ? as 2.4 for NB could be statistically different from 0.46 with Clof-, and 0.89 with Clo+. Same question for the second part of the line
For point 2: Because no interaction was observed between clofibrate and other dietary treatments, only the main effects (clofibrate and medium-chain fatty acid) were reported in the table 1. Therefore, the first p-value is for the results from clofibrate main effect, and the second p-value is for the medium-chain fatty acid main effect. To clarify this, we have stressed this point in our revised manuscript and added explanation in the opening footnote to table 1. We also declared that we presented significant results of main factors (clofibrate and dietary medium fatty acids) when the interaction (clofibrate x dietary medium fatty acids) was not statistical significant in the section of 4.7, statistical analysis. Thank you.
Table 2 : Glysuc is glysuc and clo+ or clo - ? same for the other results
For point 3: Yes. We have corrected the typing error and added the explanation to the table legend for clof- and clof+. Thanks for catching this mistake. Similar to table 1, we also stressed that no interaction was found between clofibrate and dietary medium-chain fatty acid and only the main effects were reported in the table 2.
Figure 2 : indicate whether feed supplementation was done with or without clo. From my point of view there should be 8 conditions, i.e for each feed supplementation +/- clo. So when not performed so, it should be cleraly stated, justified and the results takenn with very cautionously in the discussion section.
For point 4: Because there was no interaction between clofibrate and dietary medium-chain fatty acid treatments (clearly stated on page 4 line 139), we present only main effects: clofibrate effect was reported in Figure 1, and dietary medium-chain fatty acid treatment effects were reported in Figure 2 and 3. To clarify this, we have modified the figure legends accordingly. As we pointed before, we stressed that we presented significant results of main factors (clofibrate and dietary medium fatty acids) when the interaction (clofibrate x dietary medium fatty acids) was not statistical significant in the section of 4.7, statistical analysis. Thank you.
Same remark for figure 3 and 4. I don’t understand all letters in graph b in figure 4 : who is who in the bar ? I suppose that A is citrate synthase activity and b propionylCoA carboxylase one ? Informations of figure 5 should be in figure 4. In figure 5 SEM are important so that it is not obvious that white bar is statistically different for the black one. There is no tendency in science : either it is statistically significant, either it is not.
For point 5: Because interactions between clofibrate and dietary medium-chain fatty acid treatments Were detected for citrate synthase and propionyl-CoA carboxylase, the 8 treatments from the combination of the two main factors (clofibrate and medium-chain fatty acid) were reported. Figure 4A is the citrate synthase activity (page 7 line 178) and 4B is the propinoyl-CoA carboxylase (page 7, line 183). Because there were no treatments (clofibrate, dietary medium-chain fatty acid and the interaction between clofibrate X dietary medium-chain fatty acid) effects on acetyl-CoA carboxylase (only, we did not include this enzyme in figure 4. Instead the main effect of clofibrate (p = 0.058, treated as tendency) on acetyl-CoA carboxylase was shown in figure 5. We accepted the comment on the tendency, so the age effect on the enzyme activity was presented in figure 5 in our revised manuscript. We indicated this in the figure 5 legend. Yes, the white bar is statistically different from the black one. Because the data from newborn were treated as a reference, the difference was indicated in the legend. We have corrected the use of tendency (or trend) in the results and discussion.
The discussion will have to be modified after that the missing elements in the results section have been included.
For point 6: The discussion has been checked carefully and accordingly after the results section were revised. Thank you.
Reviewer 3 Report
Xi Lin et al have submitted the manuscript “Effects of dietary anaplerotic and ketogenic energy sources on renal fatty acid oxidation induced by clofibrate in suckling neonatal pigs.” The paper seeks to address the role of dietary anaplerosis and ketogenesis in the kidney under oxidative stress. The authors hypothesized that modification of the capacities of the CAC and ketogenic pathways affects the elevated fatty acid oxidative flux induced by clofibrate and alters the distribution of the fatty acid oxidative metabolites for renal energy generation. The authors perform a series of studies, which suggest that dietary supplementation of MCT as anaplerotic and ketogenic carbon sources may modify the metabolic pathways of renal fatty acid oxidation. However, the study did not answer clinical and pharmacological questions. It is difficult to judge whether clofibrate concentration is appropriate because their administration of clofibrate did not affect genes associated with renal fatty acid metabolism (PPARα, MCD, KCoA, PGC1α, FGF21, MCAD, LCAD andVLCAD) . They may consider the difference in its concentration, levels of plasma triglycerideor levels of proteins associated with renal fatty acid metabolism. In other words, it is not possible to judge whether the observed is due to lipid lowering or PPARαpathway. Please try to make the hypothesis clear using appropriate dose of clofibrate.
Author Response
Xi Lin et al have submitted the manuscript “Effects of dietary anaplerotic and ketogenic energy sources on renal fatty acid oxidation induced by clofibrate in suckling neonatal pigs.” The paper seeks to address the role of dietary anaplerosis and ketogenesis in the kidney under oxidative stress. The authors hypothesized that modification of the capacities of the CAC and ketogenic pathways affects the elevated fatty acid oxidative flux induced by clofibrate and alters the distribution of the fatty acid oxidative metabolites for renal energy generation. The authors perform a series of studies, which suggest that dietary supplementation of MCT as anaplerotic and ketogenic carbon sources may modify the metabolic pathways of renal fatty acid oxidation. However, the study did not answer clinical and pharmacological questions. It is difficult to judge whether clofibrate concentration is appropriate because their administration of clofibrate did not affect genes associated with renal fatty acid metabolism (PPARα, MCD, KCoA, PGC1α, FGF21, MCAD, LCAD and VLCAD). They may consider the difference in its concentration, levels of plasma triglycerideor levels of proteins associated with renal fatty acid metabolism. In other words, it is not possible to judge whether the observed is due to lipid lowering or PPARαpathway. Please try to make the hypothesis clear using appropriate dose of clofibrate.
Response: This is a very good question. The concentration of clofibrate used in this study was selected based upon our previous study (Peffer, et al., Am J Physiol Regul Integr Comp Physiol. 2005) and the concentration was confirmed to have effects on expression of genes associated with fatty acid oxidation in liver. Based on our knowledge, however, the genes listed above have never been exanimated in kidney of neonatal pigs receiving clofibrate. We do not know if this differential response is due to clofibrate concentration or to tissue differences. It has been reported that the responses of pigs and humans, to clofibrate are not same as rodent species in many studies. Currently, we are planning an experiment for evaluating this further. Certainly, in future work we will evaluate plasma triglyceride levels of proteins associated with the lipid metabolism also. Because this study was not design to answer any clinical and pharmacological questions, we have stressed our idea and the clofibrate dose used in this study in our hypothesis based on the reviewer’s critique. Thank you.
Round 2
Reviewer 2 Report
The missing informations are now available so that it is now suitable for publication from my point of view.
Author Response
Thank you for the comments!
Reviewer 3 Report
Thank you for answering my questions, and I understand your plans and situation. However, the dose setting of clofibrate is very important because clofibrate has the potential of being both a poison and a medicine in oxidative stress. I think the concentration of fibrate should be shared in any animal species.
Author Response
Point: Thank you for answering my questions, and I understand your plans and situation. However, the dose setting of clofibrate is very important because clofibrate has the potential of being both a poison and a medicine in oxidative stress. I think the concentration of fibrate should be shared in any animal species.
Response: Yes. We totally agreed. We have revised the introduction (page 2, line 55-59, line 71-75 and lines 91-93) and material method (page 10, lines 321-324) to clarify and address our focus and the dose setting of clofibarte. We also modified the conclusion slightly to improve the accuracy. Thank you for the nice comments.
Round 3
Reviewer 3 Report
Thank you for understanding my points. I would like you to continue your works for developing the field of fibrate on the kidney and oxidative stress.